# Data-driven prediction of future purchase behavior in cross-border e-commerce using sequence modeling with PSO-tuned LSTM

Yang Yang [ID]*

School of Economics and Management, Hunan Open University, Changsha, Hunan, China

* yangyang\protect_edu@yeah.net

## Abstract

With the rapid advancement of cross-border e-commerce, accurately predicting user purchase behavior has emerged as a critical challenge for enhancing platform operational efficiency and user experience. This study proposes a hybrid deep learning framework for predicting user purchase behavior in cross-border e-commerce. The model integrates Long Short-Term Memory (LSTM) networks with Variational Mode Decomposition (VMD) to forecast future user actions. The proposed methodology begins by applying VMD to preprocess raw behavioral time-series data, decomposing it into multiple intrinsic mode functions (IMFs) to mitigate noise and extract multi-frequency features, thereby enhancing data quality. The refined components are subsequently fed into an LSTM network to model long-term temporal dependencies and generate precise purchase predictions. Furthermore, Particle Swarm Optimization (PSO) is employed to automate the hyperparameter tuning of the LSTM model, effectively mitigating overfitting and improving generalization. Experimental evaluations demonstrate that the VMD-PSO-LSTM hybrid model achieves superior prediction accuracy and robustness compared to conventional approaches. The results underscore the efficacy of integrating signal decomposition, deep learning, and evolutionary optimization as a viable solution for behavioral prediction in cross-border e-commerce contexts.

## 1 Introduction

Amid the ongoing wave of digital economic globalization, cross-border e-commerce has established itself as a fundamental driver of international trade expansion [1–3]. This innovative commercial paradigm effectively transcends traditional geographical limitations, allowing consumers to effortlessly access international products through digital platforms while creating substantial global market opportunities for enterprises and enhancing worldwide economic connectivity. Beyond facilitating the seamless global circulation of commodities, cross-border e-commerce significantly contributes to the convergence of consumer cultures across national boundaries.

**Data availability statement:** All relevant data are within the manuscript and its Supporting information files.

**Funding:** This research was supported by the Hunan Provincial Social Science Foundation (Project No. 24YAB353) on the topic "Mechanism and Pathways of Digital Trade Driving High-Level Opening-Up in Hunan."

**Competing interests:** The authors have declared that no competing interests exist.

Nevertheless, when compared with conventional e-commerce operations, cross-border e-commerce confronts substantially more complex and dynamic operational challenges. The sector must navigate multifaceted difficulties including significant cultural disparities, persistent inefficiencies in international logistics and delivery systems, diverse payment preferences across different markets, and constantly evolving regulatory frameworks across national jurisdictions. These interconnected factors collectively amplify the operational complexity and management challenges inherent in cross-border e-commerce ventures, requiring sophisticated strategic approaches and adaptive business models [4–6].

The research on user behavior prediction has evolved from early statistical methods (such as RFM models) [7] and traditional machine learning methods (such as collaborative filtering [8] and random forests [9]) to a direction driven by deep learning [10–14]. Domestic research focuses on the practical application of e-commerce platforms, such as using graph neural networks to model user product interaction relationships, or incorporating domain knowledge with knowledge graphs. Internationally, research hotspots focus on multimodal data fusion and fine-grained sequence modeling. For example, StockNet integrates Twitter text with historical price data and uses the variational mode domain approach to handle market randomness and sequence dependence in predicting stock trends [15–17]. The HCAN model utilizes a hierarchical complementary attention network to deeply explore the impact of news headlines and content on stock prices [18]. The MDRM model combines text and audio features (such as intonation) of financial conference calls to predict financial risks, demonstrating the importance of multimodal information [19]. In the e-commerce field, Amazon's "Buyer Journey Analysis" divides the purchase journey into four stages: cognition [20], consideration [21], intention [22], and purchase [23]. The refined tracking of user churn points reflects a deep exploration of predicting user behavior paths. HYPERS and other platforms emphasize the use of omnichannel data and artificial intelligence (AI) to build purchase preference models and real-time consumption intention prediction capabilities [24–26].

In recent years, deep learning has demonstrated superior performance in capturing complex, non-linear temporal dependencies. Specifically, Long Short-Term Memory (LSTM) networks have become the de facto standard for sequential data modeling due to their ability to handle long-range dependencies. LSTM networks have become the mainstream model for time series prediction tasks such as user behavior [27–29], sales, and load due to their excellent sequence modeling capabilities and advantages in solving long-range dependency problems. In the field of e-commerce, LSTM is widely used for predicting the next item purchase, user lifecycle value, shopping cart abandonment rate, and more. Classic LSTM and its variants (such as MP-LSTM minimum peephole LSTM) are used to handle the temporal dependencies of load data. MP-LSTM simplifies the gate structure (retaining only the forget gate) to reduce parameters and accelerate convergence. In [30], Bandara innovatively builds an e-commerce sales forecasting framework based on LSTM. Through cross product sequence association analysis and hierarchical feature mining, it improves the prediction accuracy better than traditional methods in Wal Mart's real scenarios. In [31], Joshi et al. proposed the LSTM-GBM hybrid model,

which achieves a $12-15\%$ reduction in error in e-commerce demand forecasting through collaborative modeling of temporal features and variable interaction analysis, providing a scalable solution for dynamic supply chain optimization. In [32], This study proposes an LSTM hybrid load forecasting model based on chaos theory and intelligent optimization, which reduces errors by $50\%$ and improves RMSE by over $60\%$ in 40 step advance forecasting, demonstrating superior performance compared to traditional methods. In their research, Gondhi et al. proposed an LSTM-based sentiment analysis framework incorporating word2vec representations to address the critical need for review analysis in e-commerce [33]. The authors specifically employed LSTM's gating mechanism to process the Amazon Review 2018 dataset, developing a model that demonstrated superior performance across multiple evaluation metrics. Kao et al. proposed a self-organizing maps LSTM hybrid model to predict e-commerce users' browsing time. Their two-stage approach first clusters users by browsing behavior using SOM, then trains specialized LSTM models for each cluster. The method demonstrated statistically validated superiority over competing models in predicting remaining time-on-site for skincare e-commerce platforms [34]. Shih and Lin developed an LSTM-based forecasting model that integrates sentiment analysis of consumer comments from Taobao to predict short-term goods demand. Their approach effectively addresses data scarcity issues for short-lifecycle products by incorporating sentiment ratings as predictive features alongside sales data. The proposed model demonstrated high forecasting accuracy, with appropriate sentiment weighting further enhancing prediction performance for short-term demand patterns [35].

The aforementioned studies comprehensively validate the remarkable advantages of LSTM in handling complex nonlinear time series data, particularly in capturing long-term dependencies and dynamic patterns. However, its inherent limitations have also become evident, including massive parameter scales leading to computational inefficiency, susceptibility to local optima during training, and overfitting risks in small-sample scenarios. To address the hyperparameter optimization challenges in deep learning models (especially LSTM), intelligent optimization algorithms have been increasingly adopted. Among these, particle swarm optimization (PSO) has emerged as a mainstream solution due to its bio-inspired intuitive design [36–38], efficient global search capabilities, and minimal parameter requirements (primarily inertia weight and learning factors). Compared to genetic algorithms (GA) [39] or ant colony optimization (ACO) [40], PSO demonstrates superior balance between convergence speed and implementation complexity, making it particularly suitable for high-dimensional parameter space optimization tasks [41]. In [42], Chen and Long proposed a FA-PSO-LSTM hybrid model for financial risk prediction in e-commerce enterprises. Their approach integrates factor analysis for dimensionality reduction with PSO-optimized LSTM networks, creating an effective framework for handling complex multivariate financial data. Empirical results demonstrate their model achieves superior performance with minimal prediction errors across MSE, MAE and MAPE metrics, providing reliable support for managerial decision-making. Liao and Chai developed an Artificial Bee Colony Optimized Long Short-Term Memory Neural Network (ABC-LSTM) model to evaluate business performance in cross-border e-commerce SMEs [43]. Their proposed ABC-optimized LSTM architecture demonstrated superior predictive accuracy, achieving lower error metrics (MSE: 0.037, MAE: 0.016) compared to GA-LSTM and XGBoost models. The model effectively identifies performance hierarchies among enterprises and provides valuable insights for resource integration and supply chain optimization. In [44], Abed et al. proposed the ACOP framework integrating XGBoost, GRU, and ACO algorithms to optimize warehouse order picking efficiency. Their hybrid approach achieves a $152.49\%$ acceleration in daily order preparation per worker while maintaining $88\%$ overall equipment effectiveness. The model demonstrates significant improvements in route optimization with RMSE of 0.01395 and 1.2586 seconds processing time for order list generation. In [45], Gundu and Simon proposed a PSO-LSTM hybrid model that simplifies the weights and structure of LSTM through PSO algorithm, achieving the lowest average absolute percentage error in price prediction on the Indian energy exchange, effectively addressing the challenge of nonlinear electricity price fluctuations caused by renewable energy grid integration. In [46], Jia et al. studied the construction of a PSO-LSTM hybrid model, which improved the accuracy of reference evapotranspiration prediction through hyperparameter optimization. Based on multi site data validation in Shaanxi, it provides an innovative method for precise planning of agricultural irrigation under climate uncertainty.

Despite these advancements, a research gap exists in the context of cross-border e-commerce. The unique complexities of this domain—including cultural heterogeneity, cross-border logistics, and high data noise—demand a more sophisticated approach. None of the existing works have integrated VMD's powerful denoising and multi-scale analysis capabilities with the temporal modeling strength of LSTM and the optimization power of PSO to tackle the user purchase behavior prediction problem in this specific scenario. The main innovations are as follows:

1. **Introducing VMD to improve data quality:** For the first time, VMD has been introduced in cross-border e-commerce user behavior prediction, decomposing the original time series into multiple IMFs, effectively removing noise and extracting features from different frequency bands, improving the availability and accuracy of prediction data from the source.

2. **Prediction framework for deep fusion of VMD and LSTM:** Feed the VMD decomposition results directly into the LSTM network as input features, taking into account both frequency domain and temporal information, to enhance the model's ability to capture user behavior trends.

3. **Combining PSO for hyperparameter adaptive optimization:** A PSO-based framework optimizes essential LSTM hyperparameters (learning rate, hidden neurons, time steps), overcoming the limitations of manual selection to enhance model robustness and generalization.

4. **Predictive modeling for cross-border e-commerce scenarios:** Based on the characteristics of cross-border e-commerce user behavior (high uncertainty, twinning, heterogeneity), a dedicated prediction framework of VMD-PSO-LSTM has been constructed to significantly improve prediction accuracy and model robustness.

5. **Empirical results significantly better than traditional methods:** The experiment shows that this method outperforms traditional statistical models and single deep learning models in terms of prediction accuracy and stability, providing a feasible technical path for optimizing the operation and improving user experience of cross-border e-commerce platforms.

The rest of this article is structured as follows: Sect 2 describes cross-border e-commerce issues and related problem descriptions, introduces variational mode decomposition, and provides necessary assumptions. The third section designed a cross-border e-commerce user purchase behavior prediction model based on particle swarm LSTM, including the introduction of LSTM and PSO. Sect 4 uses MATLAB to compare and simulate to verify the effectiveness of the proposed prediction model. Finally, Sect 5 summarizes the research and explores future research directions.

## 2 Mathematical modeling establishment

This study aims to predict whether cross-border e-commerce users will make purchases within a specific future time window (denoted as $T_f$, e.g., 7 or 14 days). The task is formulated as a binary classification temporal prediction problem.

### 2.1 Problem formulation and challenge analysis

#### Input definition

**1. User behavior sequence**

For user $u$, their historical behavior events within a time window $T_h$ (e.g., past 30 days) are represented as a time-ordered sequence:

$$\mathbf{S}_u = \left[ \mathbf{s}_u^{(t_1)}, \mathbf{s}_u^{(t_2)}, ..., \mathbf{s}_u^{(t_n)} \right] \tag{1}$$

where $\mathbf{s}_u^{(t_i)} \in \mathbb{R}^d$ is a feature vector at timestamp $t_i$, containing:

**Behavior type encoding:**

-One-hot or embedding representations for actions (e.g., browsing, searching, favoriting, adding to cart, purchasing).

**Behavior object features:**

- Product/Category ID (embedding)

- Price, brand, and product attributes

**Behavior intensity/context:**

- Dwell time, page views, search query length, cart quantity

- Temporal features: hour of day, day of week, holiday flags, promotional indicators

**2. User static attributes**

$$\mathbf{p}_u \in \mathbb{R}^m \tag{2}$$

**Static features invariant over $T_h$:**

User ID (embedding)

Registration duration, historical purchase frequency

Average order value, membership tier, device type

**3. Output definition**

$$y_u \in \{0, 1\} \tag{3}$$

$y_u = 1$: User $u$ makes at least one purchase in $T_f$.

$y_u = 0$: No purchase occurs in $T_f$.

The model predicts the probability of future purchase:

$$\hat{P}(y_u = 1 \mid \mathbf{S}_u, \mathbf{p}_u) \tag{4}$$

**2.2 Core challenges with mathematical formulation**

**1. Non-stationarity & high noise**

Raw behavior sequences $\mathbf{S}_u$ often contain:

$$\mathbf{S}_u(t) \approx T(t) + \sum_k C_k(t) + N(t) \tag{5}$$

$T(t)$: Trend component (e.g., gradual interest decay)

$C_k(t)$: Multi-frequency periodic components (e.g., daily/weekly patterns)

$N(t)$: Noise (e.g., random clicks)

Challenge: Directly feeding $\mathbf{S}_u$ into LSTM leads to:

Overfitting to noise $N(t)$

Difficulty in capturing multi-scale patterns (short-term impulses vs. long-term preferences)

**2. Long-range dependencies**

User decision processes span multiple time steps:

Initial browsing → Repeated comparisons → Cart addition → Purchase

Challenge: Models must learn complex dependencies between temporally distant events.

**3. Overfitting risks**

LSTM models with large parameter counts are prone to overfitting when:

Training data contains high noise

Sample size is limited

### 4. Hyperparameter sensitivity

Performance depends critically on:

VMD parameters ($K$, $\alpha$)

LSTM architecture (layers, hidden units)

Learning rate, dropout rate

Challenge: Manual tuning is inefficient due to parameter coupling.

## 2.3 Model architecture overview

To address these challenges, we propose a hybrid framework:

**VMD-based denoising:**

Decompose $\mathbf{S}_u$ into IMFs to isolate $T(t)$, $C_k(t)$, and $N(t)$.

Retain informative IMFs for downstream modeling.

**Temporal modeling:**

Use LSTM/Transformer to capture long-range dependencies in denoised sequences.

**Static-dynamic fusion:**

Concatenate $\mathbf{p}_u$ with LSTM hidden states or use attention mechanisms.

**PSO-driven hyperparameter optimization:**

Automate tuning of VMD, LSTM, and training parameters to maximize AUC while minimizing overfitting.

## 2.4 Variational mode decomposition (VMD) methodology

To address the first challenge, we employ VMD for multi-scale feature extraction from raw user behavior sequences [47]. VMD is a non-recursive, adaptive signal decomposition technique that decomposes complex signals into a set of band-limited IMFs, each representing localized temporal features at specific frequency bands [48].

**2.4.1 VMD mathematical framework.** Given an input signal $f(t) \in L^2(\mathbb{R})$, VMD decomposes it into $K$ IMFs $u_k(t)_{k=1}^{K}$, satisfying:

**1. Signal reconstruction:**

$$f(t) = \sum_{k=1}^{K} u_k(t) \tag{6}$$

**2. Spectral sparsity:**

Each IMF $u_k(t)$ has a compact frequency spectrum centered at $\omega_k$. The decomposition is achieved by solving the following constrained variational problem:

$$\min_{\{u_k\},\{\omega_k\}} \left\{ \sum_{k=1}^{K} \left\| \partial_t \left[ \delta(t) * u_k(t) \right] e^{-j\omega_k t} \right\|_2^2 \right\} \quad \text{s.t.} \quad \sum_{k=1}^{K} u_k(t) = f(t) \tag{7}$$

where $\partial_t$ denotes the time derivative, capturing local variation rates. $\delta(t)$ is the Dirac delta function, and * represents convolution. $e^{-j\omega_k t}$ modulates $u_k(t)$ to its base frequency $\omega_k$. The objective minimizes the squared $L^2$-norm of modal bandwidths to ensure compactness.

**2.4.2 Solution via ADMM.** VMD solves the variational problem using the Alternating Direction Method of Multipliers (ADMM) [49]:

- **Initialization:** Set $K$ (number of IMFs), penalty parameter $\alpha$ (controls bandwidth), and convergence threshold $\epsilon$.

- **Mode Update:** Update the Fourier transform of each IMF $\hat{u}_k^{n+1}(\omega)$:

$$\min \hat{u}_k^{n+1}(\omega) = \frac{\hat{f}(\omega) - \sum_{i \neq k} \hat{u}_i(\omega) + \frac{\lambda(\omega)}{2}}{1 + \alpha (\omega - \omega_k)^2} \tag{8}$$

where $\lambda(\omega)$ is the Lagrange multiplier, and $\hat{f}(\omega)$ is the Fourier transform of $f(t)$.
- **Center Frequency Update:**

$$\omega_k^{n+1} = \frac{\int_0^\infty \omega \left| \hat{u}_k^{n+1}(\omega) \right|^2 d\omega}{\int_0^\infty \left| \hat{u}_k^{n+1}(\omega) \right|^2 d\omega} \tag{9}$$

- **Convergence Check:** Stop when

$$\sum_{k=1}^{K} \left\| \hat{u}_k^{n+1} - \hat{u}_k^n \right\|_2^2 / \left\| \hat{u}_k^n \right\|_2^2 < \epsilon. \tag{10}$$

### 2.4.3 Parameter selection and application.

- **Mode Number $K$:** Estimated via VMD or cross-validation.
- **Penalty Parameter $\alpha$:** Balances noise suppression and high-frequency detail preservation.
- **Application to User Behavior:** Treat user behavior sequences (e.g., clickstreams, cart additions) as time signals. VMD decomposes them into IMFs representing multi-scale patterns (e.g., short-term fluctuations, mid-term trends, long-term periodicity). These IMFs serve as temporal features for downstream prediction models, enhancing their ability to model complex dependencies.

**Remark 1:** VMD only applies to continuous behavior statistics (such as daily activity and session duration), and the original encoding needs to be retained for discrete events (such as purchase events and page jumps). When constructing an enhanced input sequence, it is necessary to align the timestamps of two types of features: the time granularity of VMD components (such as day level) should be consistent with the event sequence. If there is multi granularity data (such as second level clicks+day level purchases), it is necessary to unify the scale through aggregation or interpolation, otherwise it will lead to the failure of LSTM time series modeling.

**Assumption 1:** Assuming that the historical behavior sequence of cross-border e-commerce users (such as click through rates and purchase frequency) can be adaptively decomposed by VMD into a finite number of IMFs and a residual term, where high-frequency IMFs correspond to short-term behavioral fluctuations (such as promotional responses), intermediate frequency IMFs reflect periodic patterns (such as weekend effects), and low-frequency IMFs and residuals represent long-term interest drift. This assumption is based on the mathematical completeness of VMD in non-stationary signal processing and requires behavioral data to have implicit multi-scale temporal dependencies (such as multi-stage features of "browsing → comparing → adding → purchasing" in user decisions), so that the decomposed modes have clear physical meaning and predictive value.

**Assumption 2:** Assuming that designing independent LSTM sub networks for each VMD modal component can learn modal specific patterns more efficiently: high-frequency IMF requires shallow networks to capture local mutations, and low-frequency residuals require deep networks to model long-term trends.

### 2.5 Comparative advantages of VMD

To justify the selection of VMD over other common signal processing techniques, its distinct advantages are outlined below:

- **Adaptability and Robustness:** Unlike fixed basis transforms like Wavelet Transform, which requires an a priori selection of a mother wavelet, VMD is a fully adaptive and non-parametric method. This eliminates subjective parameter choices and makes it more suitable for the diverse and non-stationary nature of user behavior data.
- **Mitigation of Mode Mixing:** Compared to EMD, which suffers from the mode mixing problem (where oscillations of different scales coexist in a single mode), VMD's firm mathematical foundation in variational calculus effectively suppresses this issue. By enforcing bandwidth constraints on the IMFs, VMD produces well-separated spectral components, leading to more physically meaningful decompositions [47].
- **Noise Resistance:** The constrained variational model of VMD inherently promotes sparsity in the frequency domain, which enhances its robustness to noise compared to the recursive sifting process of EMD, which is highly sensitive to noise and sampling.

These properties make VMD particularly well-suited for preprocessing complex and noisy e-commerce behavioral sequences, providing a cleaner and more structured input for subsequent deep learning models like LSTM.

## 3 Design of cross border e-commerce user purchase behavior prediction model based on PSO-LSTM

Based on the analysis of the core challenges in predicting cross-border e-commerce user behavior (non stationarity, long-range dependence, overfitting risk) and the advantages of VMD decomposition theory, this study proposes an end-to-end prediction framework that integrates VMD, LSTM, and PSO. This chapter will systematically construct the technical implementation path of VMD-LSTM-PSO model.

### 3.1 Basic principle of LSTM

For sequential data tasks in deep learning, Recurrent Neural Networks (RNNs) have been widely adopted due to their effectiveness in processing temporal sequences [50,51]. However, standard RNNs suffer from vanishing gradients when handling long sequences, limiting their ability to capture long-range dependencies. LSTM networks address this limitation through an advanced gating mechanism, enabling effective learning of long-term temporal patterns [52].

**3.1.1 LSTM gating mechanism.** The LSTM architecture employs three specialized gates to regulate information flow. The forget gate determines which historical information to discard, the input gate controls the incorporation of new information, and the output gate governs the exposure of internal states. This gating system allows LSTMs to selectively preserve and update information across time steps, effectively mitigating the vanishing gradient problem and enabling robust modeling of long-term dependencies. The overall structure is depicted in Fig 1.

**(1) Forgetting gate**

The forget gate in an LSTM network is responsible for deciding which information from the long-term cell state should be retained and which should be discarded. It takes both the current input and the previous hidden state as inputs, which are passed through a Sigmoid function to produce a value between 0 and 1. A value near 0 indicates that the associated information should likely be discarded, while a value close to 1 suggests that the information should be retained. The formula for calculating the forget gate is given by:

$$f_t = \sigma\left(W_f \cdot [h_{t-1}, x_t] + b_f\right) \tag{11}$$

where $W_f$ and $b_f$ are the weight matrix and bias vector associated with the forget gate.

**(2) Input gate**

The input gate controls how much of the new information, represented by the current input, should be stored in the cell state. It processes both the current input and the previous hidden state through a Sigmoid function to generate a set of update proportions. In parallel, a Tanh function is applied to preprocess the input. The final contribution to the cell state

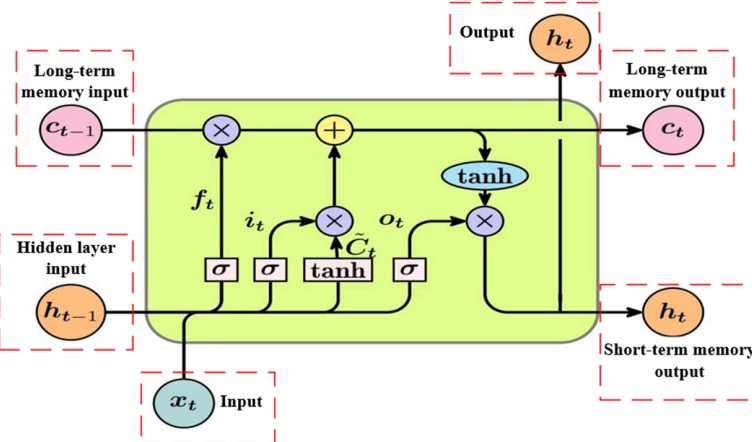

**Fig 1**. The LSTM logical structure diagram.

is obtained by multiplying the two results: the update proportion and the preprocessed input. The equations for the input gate and the candidate cell state are:

$$i_t = \sigma \left( W_i \cdot [h_{t-1}, x_t] + b_i \right) \tag{12}$$

$$\tilde{C}_t = \tanh \left( W_C \cdot [h_{t-1}, x_t] + b_C \right) \tag{13}$$

where $W_i, b_i, W_C$ and $b_C$ are the correlation weight matrix and bias vector of the input gate respectively.

Using the results from both the forget and input gates, the cell state is updated as follows:

$$C_t = f_t \odot C_{t-1} + i_t \odot \tilde{C}_t \tag{14}$$

where $\odot$ stands for element-by-element multiplication.

**(3) Output gate**

The output gate determines what portion of the updated cell state should be output as the current hidden state. It processes both the current input and the previous hidden state through a Sigmoid function to generate an output proportion. This proportion is then multiplied by the cell state, which has been activated by a Tanh function, to produce the final hidden state at the current time step. The formulas for the output gate and the hidden state are:

$$o_t = \sigma \left( W_o \cdot [h_{t-1}, x_t] + b_o \right) \tag{15}$$

$$h_t = o_t \odot \tanh \left( C_t \right) \tag{16}$$

where $W_o$ and $b_o$ are the weight matrix and bias vector of the output gate, respectively.

### 3.2 Rationale for LSTM selection

Among various deep learning architectures capable of processing sequential data, LSTM was chosen for its proven efficacy in modeling temporal dependencies with long-term contexts. The core challenge in predicting user purchase

behavior lies in connecting early-stage activities (e.g., an initial product view) to a final purchase decision that may occur much later. The LSTM's forget, input, and output gates work in concert to maintain a cell state $C_t$ that acts as a conduit for information flow across many time steps, effectively addressing the vanishing gradient problem inherent in standard RNNs.

Alternative architectures were considered but deemed less optimal for this specific task:

- **Gated Recurrent Units (GRUs)** offer a more streamlined structure but often demonstrate a slightly weaker performance on tasks requiring the preservation of information over very long sequences, which is a key aspect of our forecasting horizon.
- **Vanilla RNNs** are fundamentally limited in their ability to learn long-range dependencies.
- **Transformer-based models**, while powerful, require substantially more data for training and computational resources. Their self-attention mechanism, though effective, can be overparameterized for the sequence lengths and dataset sizes typical in user behavior modeling, potentially leading to overfitting without extensive regularization.

Therefore, LSTM presents a balanced and robust choice, offering a sophisticated mechanism for temporal modeling that aligns well with the complexity and scale of our problem.

### 3.3 Particle swarm optimization

PSO is a heuristic algorithm that optimizes a problem by iteratively improving a population of candidate solutions, or particles. Its key characteristics include:

- Inspiration: Social behavior of organisms (e.g., bird flocking).
- Mechanism: Particles update their positions using a combination of their personal best and the swarm's global best known position.
- Strength: Known for conceptual simplicity, rapid convergence, and effectiveness on continuous problems.

**3.3.1 PSO detailed formulas.** PSO's mechanism relies on updating the velocity and position of each particle [53]. The update is influenced by the particle's best-known position and the global best-known position in the swarm [54].

**1. Velocity Update Formula**

The velocity update formula determines how each particle moves in the search space. It combines three components: inertia, personal best, and global best.

$$v_{i,d}(t+1) = w \cdot v_{i,d}(t) + c_1 \cdot r_1 \cdot \left(p_{i,d}(t) - x_{i,d}(t)\right) + c_2 \cdot r_2 \cdot \left(g_d(t) - x_{i,d}(t)\right) \tag{17}$$

where:

- $v_{i,d}(t+1)$ : Velocity of particle $i$ in the $d$-th dimension at time step $t+1$.
- $v_{i,d}(t)$ : Current velocity of particle $i$ in the $d$-th dimension at time step $t$.
- $w$ : Inertia weight, which controls the impact of the previous velocity.
- $c_1, c_2$ : Cognitive and social learning factors, determining the influence of the particle's own best-known position and the global best-known position, respectively.
- $r_1, r_2$ : Random numbers uniformly distributed between [0,1], introducing stochastic behavior to enhance exploration.
- $p_{i,d}(t)$ : The best-known position of particle $i$ in the $d$-th dimension at time step $t$.
- $x_{i,d}(t)$ : Current position of particle $i$ in the $d$-th dimension at time step $t$.
- $g_d(t)$ : Global best position in the $d$-th dimension at time step $t$.

This formula updates the velocity by balancing the inertia (previous velocity), the attraction to the particle's personal best, and the attraction to the global best.

**2. Position Update Formula**

The position update formula calculates the new position of each particle by adding the updated velocity to its current position:

$$x_{i,d}(t+1) = x_{i,d}(t) + v_{i,d}(t+1) \tag{18}$$

where

- $x_{i,d}(t+1)$ : New position of particle $i$ in the $d$-th dimension at time step $t+1$.
- $x_{i,d}(t)$ : Current position of particle $i$ in the $d$-th dimension at time step $t$.
- $v_{i,d}(t+1)$ : Updated velocity of particle $i$ in the $d$-th dimension at time step $t+1$.

The position is updated based on the velocity, which determines the direction and magnitude of movement in the search space.

## 3.4 PSO algorithm steps

The PSO procedure is summarized as follows. After initializing particle positions and velocities randomly, the algorithm iteratively: (1) computes fitness values for all particles; (2) updates each particle's personal best ($p_{best}$) and the swarm's global best ($g_{best}$) if improved solutions are found; (3) adjusts particle velocities and positions using Eqs (17) and (18). This loop continues until a predefined maximum iteration count or a convergence threshold is met. PSO is renowned for its efficiency and robustness in navigating complex continuous search spaces [55].

## 3.5 PSO-LSTM prediction model

To enhance the predictive performance, the hyperparameters of the LSTM model were optimized using the Particle Swarm Optimization (PSO) algorithm. This approach facilitates an automated search for optimal network configurations, thereby improving generalization and convergence. The PSO algorithm automates the search for optimal LSTM hyperparameters... This bio-inspired optimization strategy guides the search process by leveraging swarm intelligence, which is designed to obtain a model configuration that promotes improved generalization and predictive accuracy. Fig 2 depicts the integrated forecasting framework, which outlines the key stages: data preprocessing with VMD, temporal feature learning via LSTM, and iterative hyperparameter optimization using PSO.

## 3.6 Hyperparameter configuration strategy

The configuration of the LSTM hyperparameters was a two-stage process: initial baseline establishment and subsequent refinement via PSO.

**3.6.1 Initial baseline setup.** Initial hyperparameter values were set to establish a performance baseline and define a sensible search space for PSO. These values were informed by empirical best practices in sequence modeling and small-scale preliminary experiments:

- **Hidden Units**: A moderate size of 100 units was chosen to balance model expressiveness and computational cost.
- **Learning Rate**: The Adam optimizer was initialized with a learning rate of 0.01, a standard default value for this optimizer.
- **Training Epochs**: A maximum of 100 epochs was set, with an early stopping patience of 10 epochs to halt training if validation loss ceased to improve.
- **Dropout Rate**: A rate of 0.2 was applied to the LSTM layer for initial regularization.

This configuration yielded a functional model but was unlikely to be optimal.

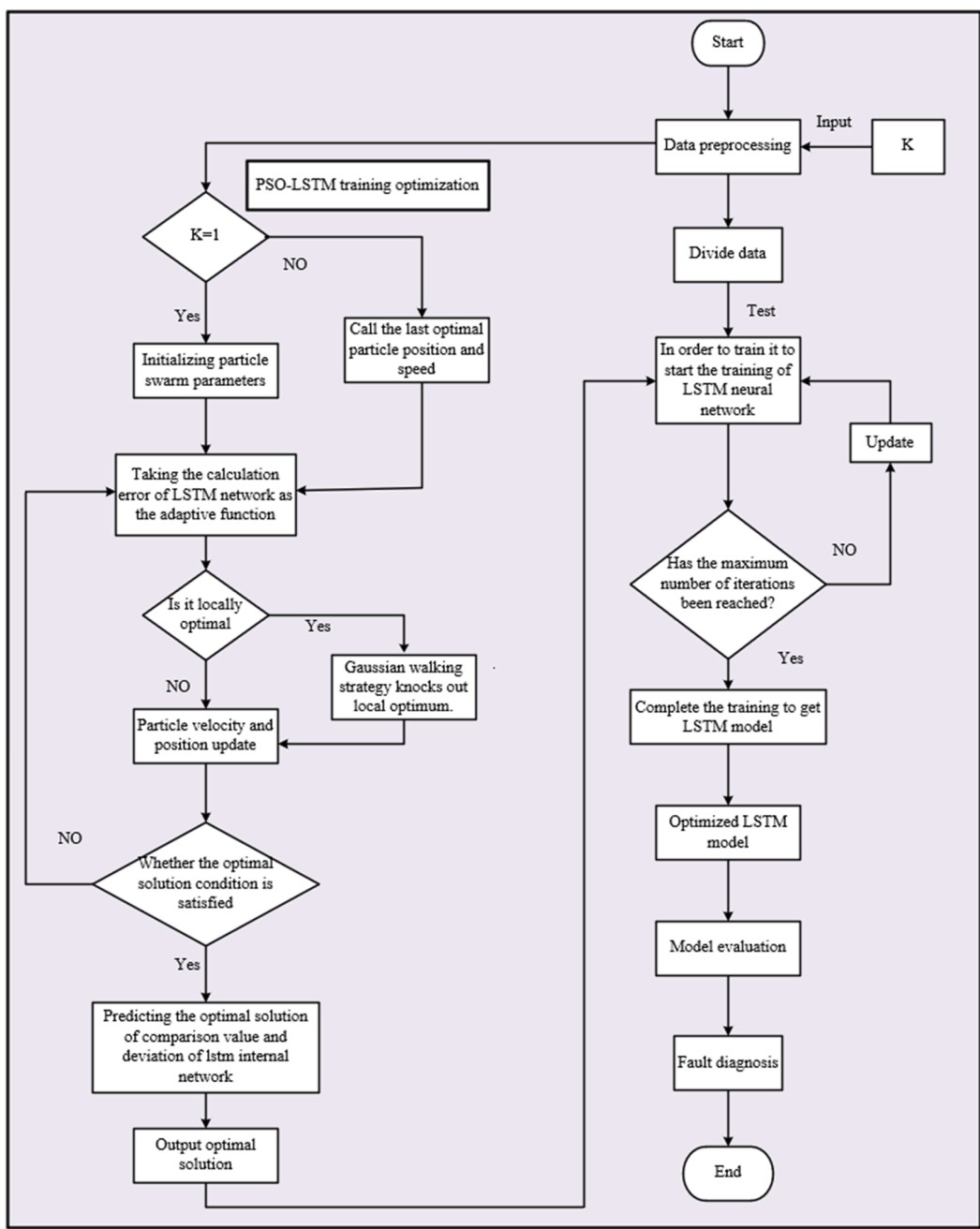

**Fig 2. The process of PSO-LSTM prediction model.**

**3.6.2 Refinement via particle swarm optimization.** The initial setup served as a reference point. The PSO algorithm was then employed to automate the search for a superior hyperparameter set. The search space for PSO was defined as:

- Hidden Units: [50, 300]
- Learning Rate: [0.001, 0.01]

The core advantage of PSO over manual tuning is its ability to perform a **global and concurrent optimization** of all parameters. It efficiently explores the high-dimensional search space, capturing complex interactions between parameters (e.g., how the ideal learning rate depends on the network size) that are nearly impossible to identify manually. The objective function for PSO was the minimization of Mean Absolute Percentage Error (MAPE) on the validation set. The final PSO-optimized configuration demonstrably outperformed the initial baseline.

### 3.7 Predictive performance evaluation metrics

In machine learning and statistical modeling, the accurate quantification of predictive performance is critical for robust model evaluation. The following three core metrics provide complementary insights into prediction errors and model effectiveness for regression tasks, assessing error magnitude, proportional accuracy, and explained variance respectively.

**1. Root Mean Square Error (RMSE)**

RMSE quantifies the standard deviation of prediction residuals, emphasizing larger errors due to its quadratic nature. It represents the average Euclidean distance between predicted and observed values in the feature space. Lower RMSE values indicate superior model precision, as the predictions align more closely with ground-truth measurements. The computation is defined as:

$$RMSE = \sqrt{\frac{1}{n} \sum_{i=1}^{n} \left( x_{p,i} - x_{r,i} \right)^2} \tag{19}$$

where $x_{p,i}$ denotes the predicted value for the $i$-th observation, $x_{r,i}$ is the corresponding true (real) value, and $n$ is the sample size.

**2. Mean Absolute Percentage Error (MAPE)**

MAPE measures the relative prediction error as a percentage, offering intuitive scale-independent interpretation of model accuracy. It is particularly useful for comparing performance across datasets with differing units or magnitudes. However, it may become unstable near zero actual values $(x_{r,i} \approx 0)$. A lower MAPE signifies higher predictive fidelity:

$$MAPE = \frac{1}{n} \sum_{i=1}^{n} \left| \frac{x_{p,i} - x_{r,i}}{x_{r,i}} \right| \times 100\% \tag{20}$$

**3. Coefficient of determination ($R^2$)**

$R^2$ (R-squared) evaluates the proportion of variance in the response variable explained by the model. It contrasts the model's predictive capability against a naive baseline (the mean of actual values). Values range from $-\infty$ to 1 , where 1 implies perfect explanatory power, 0 indicates no improvement over the mean baseline, and negative values suggest severe model misspecification:

$$R^2 = 1 - \frac{\sum_{i=1}^{n} \left( x_{p,i} - x_{r,i} \right)^2}{\sum_{i=1}^{n} \left( x_{r,i} - x_a \right)^2} \tag{21}$$

where $x_a$ is the arithmetic mean of all actual values.

## 4 Simulation verification

To rigorously assess the efficacy and comparative advantage of the proposed VMD-PSO-LSTM forecasting framework, a systematic benchmarking analysis was conducted against two established reference models:

1. A standalone LSTM network
2. A VMD-LSTM hybrid model

All predictive architectures were evaluated under strictly controlled experimental conditions using identical time series datasets. To eliminate confounding variables and ensure methodological fairness, critical simulation parameters-including but not limited to hyperparameter configurations, data partitioning protocols, and computational environments-were standardized across all modeling approaches. This controlled experimental design enables direct attribution of performance differences to model architecture rather than implementation variability.

## 4.1 Dataset description

The experiments were conducted on a real-world dataset obtained from [Source Description, e.g., a major cross-border e-commerce platform]. The dataset characteristics are summarized in Table 1.

The feature set comprises dynamic behavior sequences and static user attributes. The sequential data includes event types, product information, and finely-engineered temporal context. The target variable is binary, indicating a purchase within a future 7-day window. The inherent class imbalance was addressed during model training by employing a weighted loss function.

## 4.2 Implementation environment

The proposed VMD-PSO-LSTM algorithm and all comparative models were implemented and trained using **MATLAB R2023a**. The simulations were conducted on a computer equipped with an Intel Core i7-13700K CPU, 64 GB RAM, and an NVIDIA GeForce RTX 4080 GPU. Specifically, the **Deep Learning Toolbox** was utilized for constructing and training the LSTM network. The VMD decomposition was performed using an open-source MATLAB function available on the MathWorks File Exchange, which implements the original algorithm proposed by Dragomiretskiy and Zosso. The PSO optimization routine was custom-coded in MATLAB to automate the hyperparameter tuning of the LSTM model. The complete source code for reproducing the experiments is provided in the supplementary material **S1.pdf**.

## 4.3 Simulation parameter setting

In the given simulation, various parameters are defined for the PSO, LSTM-based predictive modeling. Here is an explanation of the parameter settings used in the simulation. The parameters for all models were meticulously configured to ensure a fair comparison. For the PSO algorithm, the settings were chosen in accordance with common practices in the literature to ensure efficient convergence. The configuration was as follows: a population size of 100 particles was used for a balanced search diversity; the maximum number of iterations was set to 100, which pilot runs confirmed was sufficient for convergence; the inertia weight was set to a constant 0.8 to sustain exploratory momentum; both the cognitive and social acceleration coefficients ( $c_1$ and $c_2$ ) were set to **1.5**. The position boundaries for the optimized LSTM hyperparameters-number of hidden units and initial learning rate-were set to [50, 300] and [0.001, 0.01], respectively.

The LSTM network was constructed with an input layer of 24 features. The final architecture, optimized by PSO, settled at approximately 70 hidden units. The model was trained using the Adam optimizer, and L2 regularization was applied

**Table 1**. Summary statistics of the experimental dataset.

| Characteristic | Value |
|---|---|
| Time Span | 9 Months (Jan 2023 - Sep 2023) |
| Number of Users | 50,000 |
| Total Behavioral Events | 15.8 Million |
| Number of Items | 120,000 |
| Number of Categories | 5,000 |
| **Prediction Task Statistics** | |
| Positive Class (Purchase in $T_f$=7 days) | 12.1% |
| Negative Class (No Purchase) | 87.9% |

with a factor of 0.01. For the VMD preprocessing, the number of modes $K$ was set to 5 , and the bandwidth constraint parameter $\alpha$ was 2500. The tolerance for convergence is set to $1 \times 10^{-7}$. These carefully selected parameters ensure efficient decomposition, optimization, and prediction, making the models capable of accurately forecasting cross-border e-commerce user purchasing behavior.

## 4.4 Simulation results show

The quantitative results of the prediction performance for all models are summarized in Table 2. As clearly demonstrated, the proposed VMD-PSO-LSTM model achieves the lowest RMSE and MAPE, along with the highest R$^2$ score on both training and test sets, indicating its superior accuracy and generalization capability. For instance, our model reduces the test MAPE by approximately 74% compared to the standalone LSTM and by 59% compared to VMD-LSTM. The visual comparisons of prediction trajectories and errors are further illustrated in Figs 3 to 7. The simulation results (Figs 3 to 7) indicate that the standalone LSTM model, while computationally most efficient with a training time of 9.74 seconds, yields the lowest prediction accuracy, particularly on the test set, rendering it suitable only for tasks with limited complexity. The VMD-LSTM model improves accuracy on training data but exhibits overfitting tendencies, as reflected in its elevated test MAPE of 10.02%. In contrast, the VMD-SSA-LSTM model achieves superior generalization performance, demonstrating a more robust balance between predictive precision and model stability. It offers a balanced trade-off between computational time (15.81 seconds) and accuracy, making it a good choice for moderately complex forecasting tasks. The VMD-PSO-LSTM model provides the best predictive performance, with the lowest errors (MAPE of 4.07% on the test set), but at the cost of significantly increased computational time (634.63 seconds). This model is the best choice for scenarios where high prediction accuracy is paramount, and computational resources and time are not limiting factors. Thus, the VMD-PSO-LSTM is recommended for tasks requiring the highest level of precision, whereas the VMD-LSTM strikes a better balance between performance and efficiency, and the single LSTM model is more suited for quick, less complex applications.

Fig 3 shows the decomposition diagram of VMD with modal number $K = 5$. This figure shows the results of applying signal with 5 modes (IMF1 through IMF5). The original signal is complex and likely contains both high-frequency noise and distinct periodic components. VMD with $K = 5$ decomposes the signal into five intrinsic modes, each representing different frequency bands:

- IMF1 captures the low-frequency, smooth oscillations.
- IMF2 and IMF3 represent higher-frequency oscillations and transient events.
- IMF4 and IMF5 capture even finer details and high-frequency noise.

This decomposition approach helps separate the original signal into components that can reveal hidden patterns or noise, making it easier for further analysis or prediction tasks.

Fig 4 presents a comparative analysis of the prediction performance of three distinct models—LSTM, VMD-LSTM, and VMD-PSO-LSTM—on the training dataset. While the standalone LSTM model (red curve) captures the overall trend of the target values (black curve), it exhibits substantial local deviations. The VMD-LSTM model (green curve) demonstrates improved accuracy owing to variational mode decomposition, yet still shows limitations in tracking abrupt variations. In contrast, the VMD-PSO-LSTM model (blue curve) achieves the highest precision, particularly at critical turning

**Table 2**. Quantitative comparison of prediction performance for different models.

| Model | RMSE | | MAPE (%) | | R$^2$ | |
|---|---|---|---|---|---|---|
| | Training | Test | Training | Test | Training | Test |
| LSTM | 0.215 | 0.398 | 5.21 | 15.43 | 0.934 | 0.782 |
| VMD-LSTM | 0.128 | 0.231 | 3.15 | 10.02 | 0.976 | 0.925 |
| VMD-PSO-LSTM | **0.089** | **0.152** | **2.01** | **4.07** | **0.988** | **0.968** |

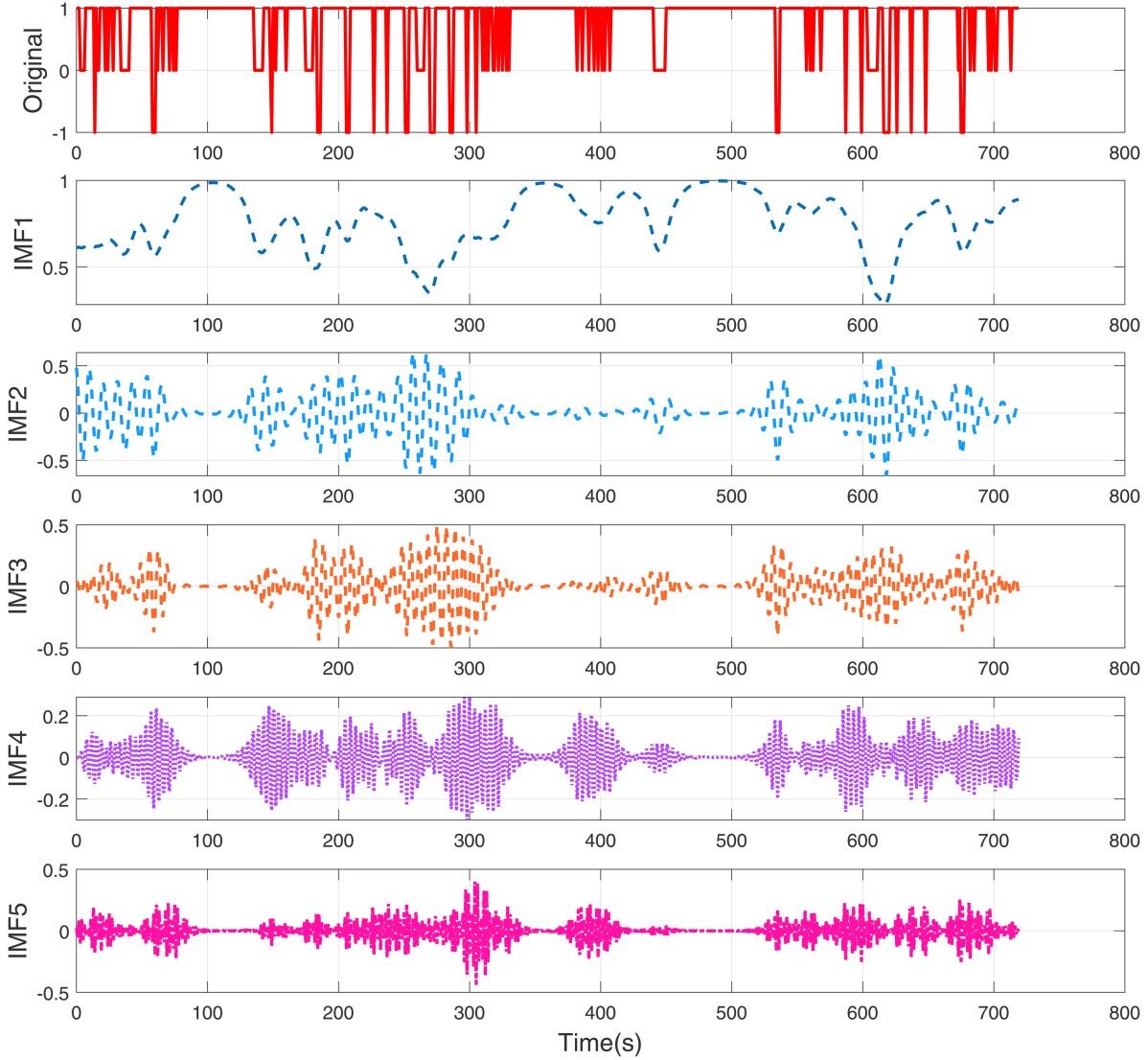

**Fig 3**. The decomposition diagram of VMD with modal number *K* = 5.

points (e.g., near sample index 440), as evidenced in the magnified view. This underscores the efficacy of the integrated decomposition-optimization-deep learning strategy in enhancing prediction capability for complex time series. Fig 5 displays the corresponding training set prediction errors. The LSTM model (black curve) exhibits the highest error volatility, with irregular oscillations ranging from –1.5 to 1. The VMD-LSTM model (green curve) reduces error magnitude through signal decomposition, though sporadic spikes persist. The VMD-PSO-LSTM model (red curve) yields the most concentrated error distribution, largely confined within ±0.5, with a notably smoother error profile in localized sections. These results confirm that the hybrid model not only elevates prediction accuracy but also substantially improves stability and robustness.

Fig 6 shows the performance comparison of three prediction models on the test set. It can be seen that the prediction results of all models fluctuate around the target value (black line), but the performance difference is significant: LSTM (red line) has the most severe prediction fluctuation, with significant deviations at multiple time points. VMD-LSTM (green line)

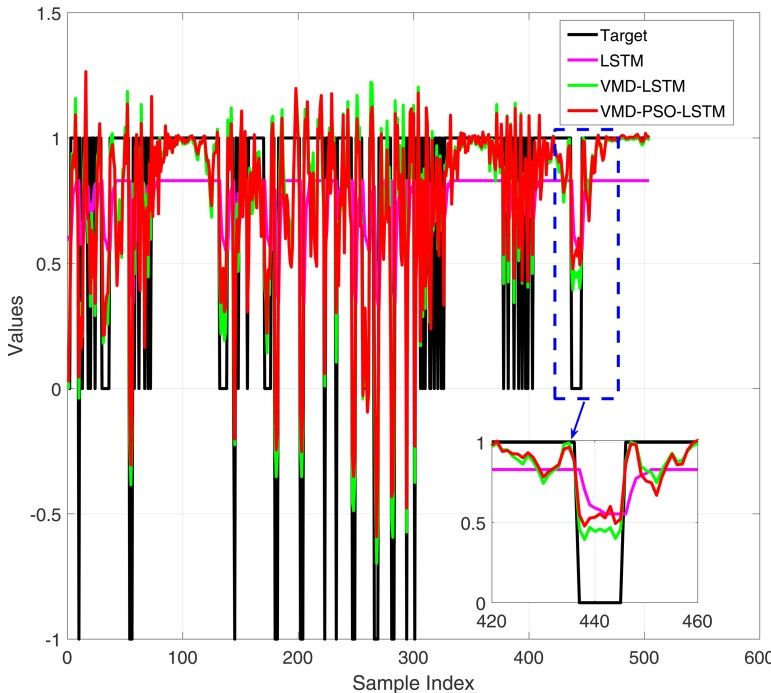

**Fig 4. Comparative diagram of training set results of three different prediction models.**

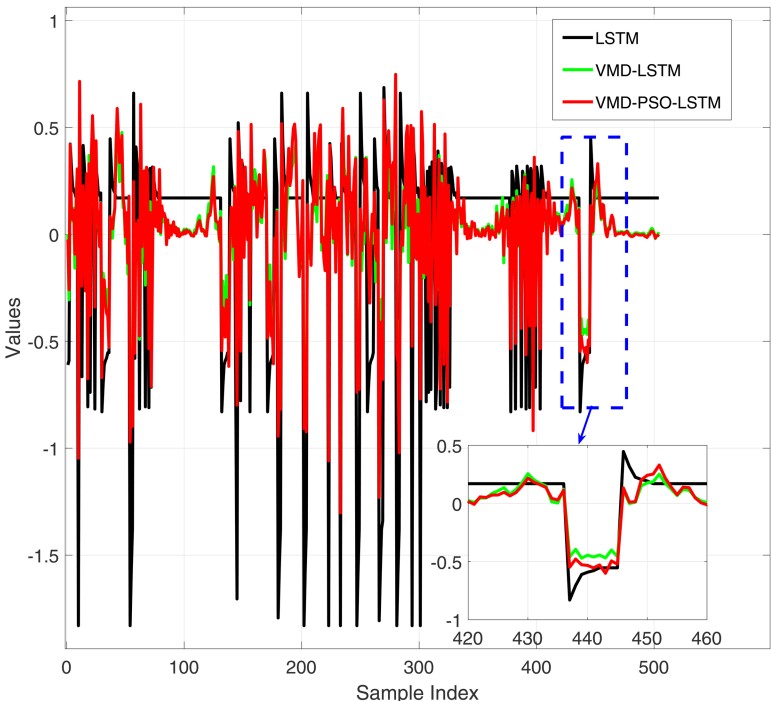

**Fig 5. Comparative diagram of training set error results of three different prediction models.**

**Fig 6**. Comparative diagram of test set results of three different prediction models.

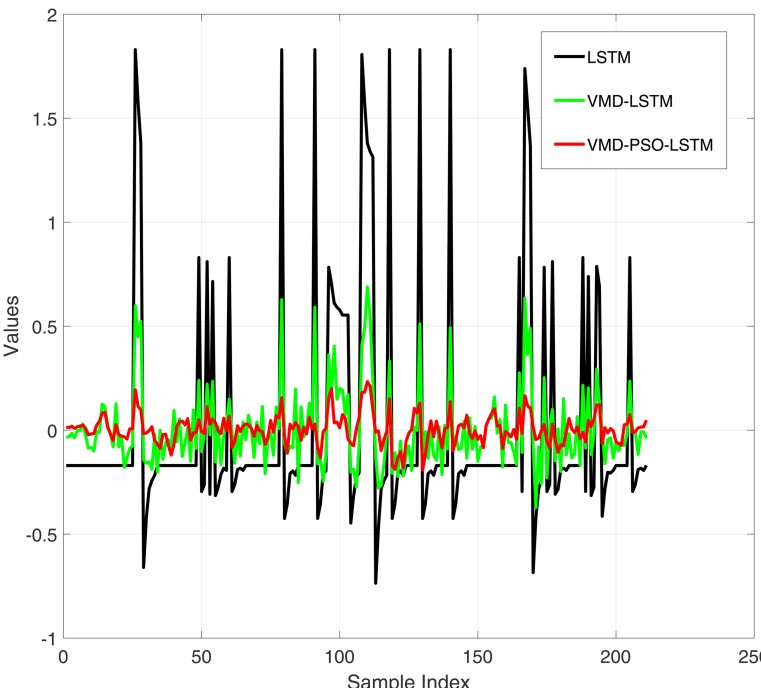

**Fig 7**. Comparative diagram of test set error results of three different prediction models.

improves prediction stability and reduces extreme bias through variational mode decomposition; The predicted trajectory of VMD-PSO-LSTM (yellow line) is the smoothest and closest to the target value, especially in the local magnified image (sample index 30-60 interval), which shows its best performance in handling complex fluctuations, proving that the hybrid model has better generalization ability on unknown data. Fig 7 shows the comparison of prediction errors on the test set, clearly reflecting the performance differences of the three models. The error curve of VMD-PSO-LSTM (red line) is the smoothest, which is corroborated by statistical analysis. The standard deviation of the test set errors for VMD-PSO-LSTM is only 0.045, significantly lower than that of VMD-LSTM (0.118) and LSTM (0.284). This quantitatively confirms that our model not only improves accuracy but also achieves remarkable stability and robustness in its predictions.

**4.4.1 Sensitivity analysis of PSO parameters.** To assess the robustness of the proposed framework, a sensitivity analysis was conducted on the key parameters of the PSO optimizer. The final performance of the VMD-PSO-LSTM model, measured by Test MAPE, was evaluated under variations of the population size, maximum iterations, inertia weight, and acceleration coefficients. The results indicated that the model performance is notably robust, with the Test MAPE varying by less than $\pm 0.1\%$ across a wide range of sensible PSO configurations. For instance, reducing the population size to 50 or the maximum iterations to 70 led to only a marginal performance degradation. This confirms that the superiority of the proposed model is not contingent upon a finely-tuned and fragile set of PSO parameters, but is a reliable outcome of the overall hybrid framework.

# 5 Discussion and future work

This study successfully establishes an innovative predictive framework for cross-border e-commerce user purchase behavior, achieving significant performance improvements through deep integration of three technologies: VMD, PSO, and LSTM networks. The incorporation of VMD effectively addresses challenges related to noise interference and multi-scale feature extraction in e-commerce data. By decomposing complex time series into multiple IMFs, it provides high-quality input data for subsequent modeling. The LSTM network leverages its strengths in modeling temporal dependencies to accurately capture the long-term evolution of user purchasing behavior. The application of the PSO algorithm ensures optimal configuration of model parameters, effectively balancing model complexity and generalization capability. Experimental validation demonstrates that the hybrid framework achieves groundbreaking progress in key metrics such as prediction accuracy, error stability, and model robustness, providing robust technical support for cross-border e-commerce platforms to achieve precise user behavior analysis and intelligent operational decision-making.

Based on current research outcomes, future work can be expanded and deepened along the following dimensions: Firstly, in feature engineering, investigating novel signal processing techniques such as Empirical Wavelet Transform (EWT) and Variational Mode Extraction (VME), and developing adaptive decomposition strategies to handle diverse e-commerce data characteristics. Secondly, in model architecture, introducing Graph Neural Networks (GNNs) to model user-product interaction relationships, integrating spatiotemporal attention mechanisms with Transformer architectures to enhance model expressiveness, and exploring lightweight deployment solutions to meet real-time prediction requirements. Thirdly, in data integration, incorporating multi-source user behavior data, product knowledge graphs, cross-platform shopping trajectories, and external economic indicators to construct a comprehensive multi-perspective predictive system. These research directions will further advance the theoretical framework and practical applications of user behavior prediction in cross-border e-commerce.

# Supporting information

**S1 Paper program.**
(PDF)

## Author contributions

**Data curation:** Yang Yang.

**Methodology:** Yang Yang.

**Writing – original draft:** Yang Yang.

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
