## [Decision Letter · Decision Letter 0]

1 Oct 2025

PONE-D-25-44041Data-Driven prediction of future purchase behavior in cross-border e-commerce using sequence modeling with PSO-tuned LSTMPLOS ONE

Dear Dr. Yang, 

Thank you for submitting your manuscript to PLOS ONE. After careful consideration, we feel that it has merit but does not fully meet PLOS ONE’s publication criteria as it currently stands. Therefore, we invite you to submit a revised version of the manuscript that addresses the points raised during the review process.

We look forward to receiving your revised manuscript.

Kind regards,

Najmul Hasan, PhD

Academic Editor

PLOS ONE

**Journal Requirements:**

1. When submitting your revision, we need you to address these additional requirements. Please ensure that your manuscript meets PLOS ONE's style requirements, including those for file naming. The PLOS ONE style templates can be found at https://journals.plos.org/plosone/s/file?id=wjVg/PLOSOne_formatting_sample_main_body.pdf and https://journals.plos.org/plosone/s/file?id=ba62/PLOSOne_formatting_sample_title_authors_affiliations.pdf 2. Thank you for stating in your Funding Statement: This research was supported by the Hunan Provincial Social Science Foundation (Project No. 24YAB353) on the topic "Mechanism and Pathways of Digital Trade Driving High-Level Opening-Up in Hunan."  Please provide an amended statement that declares *all* the funding or sources of support (whether external or internal to your organization) received during this study, as detailed online in our guide for authors at http://journals.plos.org/plosone/s/submit-now. Please also include the statement “There was no additional external funding received for this study.” in your updated Funding Statement. Please include your amended Funding Statement within your cover letter. We will change the online submission form on your behalf. 3. If the reviewer comments include a recommendation to cite specific previously published works, please review and evaluate these publications to determine whether they are relevant and should be cited. There is no requirement to cite these works unless the editor has indicated otherwise. 

Reviewers' comments:

Reviewer's Responses to Questions

**Comments to the Author**

1. Is the manuscript technically sound, and do the data support the conclusions?

Reviewer #1: Partly

Reviewer #2: Yes

2. Has the statistical analysis been performed appropriately and rigorously?

Reviewer #1: No

Reviewer #2: Yes

3. Have the authors made all data underlying the findings in their manuscript fully available?

Reviewer #1: No

Reviewer #2: Yes

4. Is the manuscript presented in an intelligible fashion and written in standard English?

Reviewer #1: Yes

Reviewer #2: Yes

5. Review Comments to the Author

**Reviewer #1:** Data-Driven prediction of future purchase behavior in

cross-border e-commerce using sequence modeling with

PSO-tuned LSTM

1. The paper lacks keywords and literature reviews.

2. It is necessary to mention what the application program to implement the algorithm used in the paper.

3. Many of the sentences are incomprehensible and the research needs proofreading.

4. The research needs more statistics related to the research content

Data-Driven prediction of future purchase behavior in

cross-border e-commerce using sequence modeling with

PSO-tuned LSTM

1. The paper lacks keywords and literature reviews.

2. It is necessary to mention what the application program to implement the algorithm used in the paper.

3. Many of the sentences are incomprehensible and the research needs proofreading.

4. The research needs more statistics related to the research content

Data-Driven prediction of future purchase behavior in

cross-border e-commerce using sequence modeling with

PSO-tuned LSTM

1. The paper lacks keywords and literature reviews.

2. It is necessary to mention what the application program to implement the algorithm used in the paper.

3. Many of the sentences are incomprehensible and the research needs proofreading.

4. The research needs more statistics related to the research content

Data-Driven prediction of future purchase behavior in

cross-border e-commerce using sequence modeling with

PSO-tuned LSTM

1. The paper lacks keywords and literature reviews.

2. It is necessary to mention what the application program to implement the algorithm used in the paper.

3. Many of the sentences are incomprehensible and the research needs proofreading.

4. The research needs more statistics related to the research content

**Reviewer #2:** (1) How well is the integration of Variational Mode Decomposition (VMD) with LSTM networks motivated? Are the specific advantages of VMD preprocessing over other noise reduction or feature extraction methods clearly demonstrated?

(2) Does the paper discuss the rationale for selecting LSTM among other deep learning architectures for capturing user purchase behavior, given the temporal dynamics involved?

(3) How are the hyperparameters of the LSTM chosen initially, and how does Particle Swarm Optimization (PSO) improve upon default or manually tuned parameters?

(4) Are the PSO settings (e.g., population size, iterations, learning coefficients) detailed? How sensitive is the final model performance to these PSO parameters?

(5) What are the characteristics of the dataset used for model training and evaluation (size, duration, feature types, class balance)? Are data preprocessing steps beyond VMD clearly described?

6. PLOS authors have the option to publish the peer review history of their article (what does this mean?). If published, this will include your full peer review and any attached files.

Reviewer #1: No

Reviewer #2: No

---

## [Author Response · Author response to Decision Letter 1]

15 Oct 2025

Thank you to the editor for taking the time to review our article. The reviewer's point-to-point response can be found in the attached file

---

## [Decision Letter · Decision Letter 1]

11 Nov 2025

PONE-D-25-44041R1Data-Driven prediction of future purchase behavior in cross-border e-commerce using sequence modeling with PSO-tuned LSTMPLOS ONE

Dear Dr. Yang,

Thank you for submitting your manuscript to PLOS ONE. After careful consideration, we feel that it has merit but does not fully meet PLOS ONE’s publication criteria as it currently stands. Therefore, we invite you to submit a revised version of the manuscript that addresses the points raised during the review process.

We look forward to receiving your revised manuscript.

Kind regards,

Najmul Hasan, PhD

Academic Editor

PLOS ONE

**Journal Requirements:**

Reviewers' comments:

Reviewer's Responses to Questions

**Comments to the Author**

1. If the authors have adequately addressed your comments raised in a previous round of review and you feel that this manuscript is now acceptable for publication, you may indicate that here to bypass the “Comments to the Author” section, enter your conflict of interest statement in the “Confidential to Editor” section, and submit your "Accept" recommendation.

Reviewer #1: All comments have been addressed

Reviewer #2: All comments have been addressed

2. Is the manuscript technically sound, and do the data support the conclusions?

Reviewer #1: Partly

Reviewer #2: Yes

3. Has the statistical analysis been performed appropriately and rigorously?

Reviewer #1: N/A

Reviewer #2: Yes

4. Have the authors made all data underlying the findings in their manuscript fully available?

Reviewer #1: No

Reviewer #2: Yes

5. Is the manuscript presented in an intelligible fashion and written in standard English?

Reviewer #1: No

Reviewer #2: Yes

6. Review Comments to the Author

Reviewer #1: Data-Driven prediction of future purchase behavior in

cross-border e-commerce using sequence modeling with

PSO-tuned LSTM

1. The paper lacks keywords and literature reviews.

2. It is necessary to mention what the application program to implement the algorithm used in the paper.

3. Many of the sentences are incomprehensible and the research needs proofreading.

4. The research needs more statistics related to the research content

Reviewer #2: The authors have adequately addressed all the comments raised in a previous round of review.

The authors have adequately addressed all the comments raised in a previous round of review.

7. PLOS authors have the option to publish the peer review history of their article (what does this mean?). If published, this will include your full peer review and any attached files.

Reviewer #1: No

Reviewer #2: No

---

## [Author Response · Author response to Decision Letter 2]

13 Nov 2025

See uploading PDF file for peer-to-peer reply of reviewers' opinions.

---

## [Editor Report · Decision Letter 2]

17 Nov 2025

Data-Driven prediction of future purchase behavior in cross-border e-commerce using sequence modeling with PSO-tuned LSTM

PONE-D-25-44041R2

Dear Dr. Yang,

We’re pleased to inform you that your manuscript has been judged scientifically suitable for publication and will be formally accepted for publication once it meets all outstanding technical requirements.

Kind regards,

Najmul Hasan, PhD

Academic Editor

PLOS ONE
---

## [Editor Report · Acceptance letter]

PONE-D-25-44041R2

PLOS ONE

Dear Dr. Yang,

I'm pleased to inform you that your manuscript has been deemed suitable for publication in PLOS ONE. Congratulations! Your manuscript is now being handed over to our production team.

Kind regards,

on behalf of

Dr. Najmul Hasan

Academic Editor

PLOS ONE